# Effect of *Strobilanthes tonkinensis* Lindau Addition on Black Tea Flavor Quality and Volatile Metabolite Content

**DOI:** 10.3390/foods11121678

**Published:** 2022-06-07

**Authors:** Jinjie Hua, Jia Li, Wen Ouyang, Jinjin Wang, Haibo Yuan, Yongwen Jiang

**Affiliations:** 1Tea Research Institute, Chinese Academy of Agricultural Sciences, Hangzhou 310008, China; huajinjie@tricaas.com (J.H.); jiali1986@tricaas.com (J.L.); ouyangwen@tricaas.com (W.O.); jinjinwangtkzc@tricaas.com (J.W.); 2Key Laboratory of Tea Quality and Safety Control, Ministry of Agriculture and Rural Affairs, Hangzhou 310008, China

**Keywords:** black tea, *Strobilanthes tonkinensis* lindau, sweet and glutinous rice flavor, gas chromatography–tandem mass spectrometry, volatile metabolites, odor activity value

## Abstract

The characteristic aroma of Chinese black tea (BT) produced in summer usually lacks intensity and persistence, reducing consumer acceptance and market demand. *Strobilanthes tonkinensis* Lindau (STL) possesses excellent biological characteristics, making it a promising novel tea ingredient. We investigated the effects of different addition methods and concentrations for the novel additive STL on the sensory quality of BT. A 20:1500 g/g addition to rolled tea leaves was identified as the best BT with STL (BoS) treatment. We identified 141 volatile metabolites (VMs) for the first time in STL, with high alcohol, ester, ketone, enyne, alkyne, and alkane contents. Partial least-squares discriminant analysis distinguished the samples and revealed 28, 26, and 14 differential VMs in STL vs. BoS, BT vs. STL, and BT vs. BoS comparisons, respectively. Using a combination of Venn, multiple experiment viewer, and odor activity value analysis, 16 key differential VMs were identified. Compared to BT, the 1-octen-3-ol, 1-hexanol, 1-dodecanol, (E)-3-hexen-1-ol, phenylethyl alcohol, and 2-methoxy-3-(2-methylpropyl)-pyrazine contents in BoS were 394.7%, 53.6%, 34.1%, 24.4%, 9.9%, and 5.7% higher, respectively. BoS combined the aromatic properties of BT and STL into a sweet and glutinous rice flavor. The results provide theoretical support for future research and development of novel BT-based products.

## 1. Introduction

Black tea (BT) is the most widely consumed tea product worldwide. It contains many functional components, such as theaflavins, theasinensins, amino acids, and flavonoids, and has important health benefits, including antimicrobial, anticancer, antioxidant, antiviral, and caries-preventive effects [1,2,3]. BT produced in China, especially in summer, lacks intensity and persistence in its characteristic aroma [4], which reduces consumer acceptance and, consequently, market demand. To solve this problem, a series of BT products with new flavors have been developed by blending raw BT with flowers from other plant species, such as jasmine, rose, and *Osmanthus* [5,6], to create new BT products with characteristic flower or fruit flavors. However, the development of these new products has predominantly been based on the use of raw BT, and systematic research on the optimal blending processes is lacking. Consequently, the quality of the new BT products is neither stable nor superior. Therefore, novel additives during BT processing are required to improve the aroma and flavor of BT and to ensure optimal blending and stable quality.

*Strobilanthes tonkinensis* Lindau (STL) is a medicinal and aromatic plant native to Yunnan Province, China, and is now extensively distributed across Thailand and Vietnam [7,8]. STL has a unique aroma of glutinous rice when dried and is rich in biologically active compounds such as amino acids and flavonoids [9,10,11] and aromatic components such as carene, benzenamine, geraniol, and linalool [7,12,13]. Moreover, STL has many biological activities, including anti-aging, antiviral, anti-obesity, and anticoagulant properties [14,15,16]. Thus, it may be a promising candidate ingredient to help modulate the flavors and fragrances of tea.

In this study, STL was organically blended with BT during BT processing. By determining the optimum addition methods and concentrations, the influence of STL on the aroma of BT was elucidated, and the best technical parameters for the processing of BT with STL (BoS) were determined. Moreover, combined with sensory quality evaluation, infrared-assisted extraction coupled to headspace solid-phase microextraction (IRAE-HS-SPME) pretreatment technology and gas chromatography–tandem mass spectrometry (GC-MS/MS) were used to determine the influence of STL on the volatile compounds and aroma of BT. Multiple experiment viewer (MEV) analysis, partial least-squares discriminant analysis (PLS-DA), Venn, and odor activity value (OAV) analysis were used to determine the key volatile substances present in STL and BoS, the influence of STL on the aroma and flavor of BT, and the key differential volatile substances. We aimed to provide sound theoretical support and innovative ideas to contribute to the future research and development of new BT products.

## 2. Materials and Methods

### 2.1. Materials and Reagents

Fresh tea shoots of ‘Fudingdabai’ (*Camellia sinensis* L.) were collected in June 2020 from the Zhejiang Kaihua tea production area. The moisture content of fresh shoots was approximately 79%. Each tea shoot consisted of two leaves and one bud. Dried STL shoots with two leaves were purchased from the Yunwei Guming Tea Company (Honghe, China).

All chromatography-grade standards were purchased from the Yuanye Biological Technology Company (Shanghai, China). The purity of the chemical standards was more than 98%.

### 2.2. Tea Processing and Sample Treatment

Tea processing and sample treatment in this article were carried out in line with published studies [17], as detailed below.

#### 2.2.1. Withering

Fresh 5-cm-thick leaves were placed in an artificial climatic incubator (PRX-500D type, Ningbo Prandt Instrument Co., Ltd., Ningbo, China) for withering at 3000 lux illumination intensity, 28 °C, and 70% relative humidity (RH) until the moisture content (weight basis) was reduced to 62–64%.

#### 2.2.2. Rolling

The withered leaves were rolled in a rolling machine (6CR-35 type, Zhejiang Chunjiang Tea Machinery Co., Ltd., Hangzhou, China) for 75 min, initially under no pressure for 25 min, and then under light pressure for 20 min, followed by heavy pressure for 10 min, light pressure for 15 min, and finally, no pressure for 5 min.

#### 2.2.3. Fermenting

The rolled leaves were fermented for 3 h in an artificial climatic room at 28 °C and 95% RH.

#### 2.2.4. Drying

The fermented leaves were dried at 110 °C in a hot-air drying machine (Zhejiang Chunjiang Tea Machinery Co., Ltd.) for approximately 15 min until 100 g of the leaves contained 20–25 g of water, and then they were spread out for 30 min. Finally, the BT samples were dried for approximately 30 min at 90 °C to reduce their moisture content to approximately 5%.

Different addition treatments and concentrations of STL were evaluated using the same BT leaf-processing conditions as described above. Details of the methods used for the addition of STL during leaf processing are shown in Table 1. All tea samples were collected after the final drying period and stored in an ultra-low temperature freezer (991 type, Thermo Fisher Scientific Inc., Waltham, MA, USA) at −80 °C until analysis.

### 2.3. Tea Sensory Quality Evaluation

Tea sensory quality was assessed according to the National Standard of China (GB/T23776-2018) by five professional tea expert panelists (three women and two men, 25–48 years of age). The 100-score tea quality grading system was used, in which 30% was assigned for the tea taste, 25% for aroma, 25% for appearance, 10% for the liquor color, and 10% for the infused leaf. Tea samples (3.0 g) were added to each corresponding teacup containing pure boiling water (150 mL) and soaked for 5 min, followed by immediate separation. The sensory properties of tea samples were evaluated in the following order: aroma for hot scent, liquor color, aroma for warm scent, taste, aroma for cooled-down scent, and infused leaves [17,18].

### 2.4. Analysis of the Volatile Metabolites

#### 2.4.1. Sample Preparation

The samples (0.5 g) were accurately weighed, and 0.0001 g was placed in headspace vials (20 mL). To this, 10 μL of 20 mg/L ethyl caprate in 1 mL of boiling pure water was added. The headspace vial cap was immediately tightened. The cap was protected by being impaled with a DVB/CAR/PDMS fiber head (Supelco, Bellefonte, PA, USA) with a manual handle (SPME, Supelco). The vial was then placed over a 100 W infrared device at a distance of 6 cm for 15 min, and the fiber head was inserted into a GC-MS inlet and desorbed at 250 °C for 2 min. Each sample was assessed three effective times for IRAE-HS-SPME [19,20,21].

#### 2.4.2. GC-MS/MS Analysis

A gas chromatograph (7890B-7000C type, Agilent Corporation, Santa Clara, CA, USA) was used in the splitless injection mode. High-purity helium (99.99%) at a 1.0 mL/min flow rate was used as the carrier gas. The GC injector temperature was 250 °C, and the program of column temperature was as follows: initial temperature of 50 °C maintained for 5 min; an increase to 150 °C at 4 °C/min, which was maintained for 2 min, and an increase to 270 °C at 10 °C/min, which was maintained for 6 min [19,21].

#### 2.4.3. Qualitative and Quantitative Analyses

Based on the NIST11 standard library, volatile metabolites (VMs) with over 80% similarity were screened using the unknown analysis program Agilent Mass Hunter [20]. Kovàts retention indices (RI) for each volatile metabolite were obtained by calculating the linear formula of *n*-alkanes (C7–C40) and by comparing them with theoretical RI having the same chromatographic column in the NIST Chemistry WebBook (http://webbook.nist.gov/chemistry/, accessed on 1 December 2020), based on the difference of RI within 30. Moreover, some key VMs, such as trans-β-ionone, α-ionone, (E)-4-heptenal, 1-hexanol, linalool, octanal, 1-decanol, naphthalene, indole, decanal, geraniol, cedrol, linoleic acid ethyl ester, and hexadecenoic acid ethyl ester, were identified with authentic standards. For absolute quantitative analysis, the mass concentrations of the VMs were calculated with the internal or authentic standard method using the following equation: *C_i_* = (*C_is_* × *A_i_*)/*A_is_*,
where *C_i_* is the mass concentration of any component (μg/g), *C_is_* is the mass concentration of the internal or authentic standard (μg/g), *A_i_* is the chromatographic peak area of any component, and *A_is_* is the chromatographic peak area of the internal or authentic standard.

#### 2.4.4. OAV Analysis

OAV is the ratio of the concentration of VMs to the odor threshold in a tea sample [20] and is calculated using the following equation: OAV = *c*/*OT*,
where *c* is the volatile compound concentration (μg/kg), and *OT* is the volatile compound odor threshold in pure water (μg/kg).

### 2.5. Statistical Analysis

The tests were repeated three times, and all experimental data were calculated as the average of three replicates and reported as means ± standard deviation. SAS 9.4 (SAS Institute Inc., Cary, NC, USA) was used to analyze the significant differences in concentrations and contents of the VMs in the different tea samples. The differential volatiles among tea samples were analyzed using SIMCA 13.0 (Umetrics, Umeå, Sweden). Multiple experiment viewer (MEV) (MEV 9.0; https://mev-tm4-org.caas.cn, accessed on 15 June 2021) analysis was used to compare the contributions of the differential VMs. The correlations of VMs were plotted using Venn graphs and R (http://www.r-project.org/, accessed on 15 September 2021) according to the previously described methods [21].

## 3. Results and Discussion

### 3.1. Effects of STL Additions on the Sensory Qualities of BT

Aroma and taste have an important impact on BT quality and its market value as they are the key attributes in tea sensory evaluation [22,23]. The different methods of STL addition and the different STL concentrations tested on the tea leaves were found to have a substantial impact on the taste and aroma of BT but little impact on the appearance, liquor color, and infused leaves (Table 2). STL leaves were found to have a strong glutinous flavor and a sweet and thick taste, which is in accordance with previous studies [11,12], and the addition of STL significantly influenced BT flavor. The comparison among the different methods of STL addition during tea leaf processing showed that the sensory quality of the BoS samples obtained using treatment #2 (i.e., adding STL to rolled tea leaves) was the best, resulting in a glutinous sweet aroma and a sweet mellow taste. In contrast, adding STL to withered tea leaves resulted in an excessively heavy glutinous flavor that concealed the characteristic aroma of the BT. Adding STL to fermented tea leaves rendered the BT aroma a little grassy, and although the taste was sweet and mellow, STL addition did not improve the sensory qualities of the BT. 

The comparisons among the different concentrations of STL tested revealed that the sensory quality of BoS samples prepared using treatment #2 (i.e., 20:1500 *w*/*w*) was the highest, with a strong and lasting glutinous sweet aroma and a sweet, mellow, and thick taste. In contrast, STL concentrations exceeding 20:1500 (*w*/*w*) concealed the characteristic flavor of BT, which did not benefit BT quality. When the concentration was too low, the aroma and taste of the STL had no positive effect on the BT flavor. To clarify the influence of STL on the properties of BT aroma, BT, BoS (as per treatment #2), and STL samples were selected for further study to detect and analyze their VMs.

### 3.2. Effects of STL Addition on BT VMs

#### 3.2.1. Identification of VMs

A total of 141 VMs were detected among the tea samples (Appendix A), including 28 alcohols, 23 esters, 12 aldehydes, 11 ketones, 16 aromatic hydrocarbons, 14 terpenes and alkynes, 32 alkanes, and 5 others. These results differed slightly when compared with those of previous studies [7,13]. This might be because of the different sample pretreatments and volatile compound detection technologies used. Using the novel approach of IRAE-HS-SPME combined with GC–MS/MS, the present study obtained more types and quantities of VMs, representing an improvement over previous studies. Among them, the most abundant compounds were (E)-2-hexenal, 1-methyl-1H-1,2,4-triazole, (E)-3-hexen-1-ol, benzaldehyde, 1-octen-3-ol, 1-phenyl-1,2-propanediol, linalool, isophorone, benzeneacetaldehyde, phenylethyl alcohol, methyl salicylate, nerol, and *trans*-β-ionone.

The total VMs content and the content and proportion of the volatile compound types were significantly (*p* < 0.05) different among the BT, BoS, and STL samples (Appendix A), with the highest total volatile compound content being in the STL samples (*p* < 0.05), followed by those in the BT and then the BoS samples, although the difference between the last two sample types was not significant. The ester, alcohol, ketone, enyne, alkyne, and alkane contents were the highest in the STL (*p* < 0.05), which is consistent with the findings of Zhang [24]. However, the aldehyde content was the highest in the BT samples (*p* < 0.05). In turn, the proportions of the esters, ketones, enynes, alkynes, and alkanes were the highest in the STL samples (*p* < 0.05), whereas those of the aromatic hydrocarbons and alcohols were the highest in the BT and STL samples, respectively (*p* < 0.05 for each). Through appropriate addition technology, STL could confer BT samples with a variety of volatile metabolites, thereby enhancing the aroma quality of BT. The VMs present in the BT and STL samples were combined in the BoS samples. Among them, alcohol and alkane contents were significantly higher in the latter than in the BT samples, which was consistent with the sensory quality evaluation results.

#### 3.2.2. MEV Analysis

MEV analysis was performed for the three tea sample types based on the 141 VMs detected (Appendix A) to identify the key compounds contributing to the STL aroma and the influence of STL on BT aroma improvement. The BT and BoS samples were found to be clustered together into one class, whereas the STL samples formed a separate category (Figure 1).

In addition, the 141 VMs detected were grouped into three classes. The first class included VMs that were the highest in STL samples, such as 3-octanol, 1-octen-3-ol, 1,7,7-trimethyl-bicyclo[2.2.1]hept-2-ene, 2-ethyl-1-hexanol, 1-phenyl-1,2-propanediol, 3,6,6-trimethyl-bicyclo[3.1.1]hept-2-ene, isophorone, 3-carene, acetophenone, 4-chloro-2-methyl-1-phenyl-3-buten-1-ol, 1-nonanol, linalool oxide (pyranoid), 3-hexenyl ester, (Z)-butanoic acid, 1-dodecanol, formic acid dodecyl ester, 6-methyl-pentadecane, 2-hydroxy-benzoic acid ethyl ester, α-ionone, and *trans*-β-ionone, which is partly consistent with the results reported by Zhang et al. [7]. Due to different pretreatment and detection methods, the present study only obtained 79 types of VMs, among which pyrimidines had the highest content; further, common VMs such as 3-octanone and 3-octanol were also found. These compounds contributed largely to the glutinous rice flavor of the STL samples. The second class included VMs that were the richest in the BT samples and jointly contributed to a distinctly sweet aroma, such as 2-methyl-butanal, 3-methyl-butanal, anhydride propanoic acid, (E)-2-hexenal, hexanal, 1-methyl-1H-1,2,4-triazole, (E)-3-hexen-1-ol, 1-hexanol, methoxy-phenyl-oxime, β-myrcene, benzeneacetaldehyde, linalool, nonanal, phenylethyl alcohol, 3,3,5-trimethyl- heptane, and nerol, which is in line with the findings of previous studies [25,26]; these VMs were the typical characteristic substances responsible for the sweet aroma of black tea, which further proved the validity and accuracy of our research. Finally, the third class included the VMs that were the richest in the BoS samples, such as dimethyl ether, 2-methyl-butanal, 3-methyl- butanal, toluene, (E)-2-hexenal, 2-methoxy-furan, (E)-3-hexen-1-ol, *cis*-2-methyl-cyclopentanol, 1-hexanol, 2-methyl-pentanoic acid anhydride, decane, linalool, phenylethyl alcohol, methyl salicylate, decanal, nerol, and (Z)-hexanoic acid-3-hexenyl ester. Moreover, the BoS samples combined the aroma properties of the BT and STL samples, resulting in a sweet and glutinous rice flavor.

#### 3.2.3. PLS-DA Analysis

PLS-DA is widely used in food and agriculture research as a discriminant analytical method [18,21,22,23]. Therefore, we applied PLS-DA to explore the influence of STL addition on VMs and aroma improvements of BT and to clarify the differences in volatile compound contents in the BT, BoS, and STL samples according to the 141 compounds found in dry tea. The three sample types were effectively distinguished by PLS-DA (Figure 2a). The STL samples clustered together in both quadrants on the left side, whereas the BoS and BT samples clustered in the upper-right and lower-right quadrants, respectively. These results were consistent with the sensory quality evaluation (Table 2) and the grouping pattern obtained using the heatmap cluster analysis (Figure 1). The model was verified by cross-validation after selecting three principal components. The related variables were as follows: R^2^X (cum) was 0.955, R^2^Y (cum) was 0.988, and Q^2^ (cum) was 0.949, which indicated that this model was reliable and had strong prediction abilities.

The loading plot (Figure 2b) clearly illustrates the differences in the volatile compound contents among the BT, STL, and BoS samples. For example, the concentrations of VMs such as hexanal, 1-methyl-1H-1,2,4-triazole, ethylbenzene, 3-octanone, *o*-cymene, benzeneacetaldehyde, linalool, isobutyl pentyl ester, and neophytadiene were significantly higher in BT (*p* < 0.05) than in the other two sample types, in which hexanal has a fruity and sweaty odor [2], and benzeneacetaldehyde and linalool impart a sweet and woody flavor [25]. Meanwhile, the concentrations of VMs such as 1-octen-3-ol, naphthalene, l-α-terpineol, 2-methoxy-3-(2-methylpropyl)-pyrazine, 5-methyl-tridecane, isophorone, *n*-pentadecanol, 1-dodecanol, and *trans*-β-ionone were the highest in the STL samples (*p* < 0.05). These accounted for glutinous rice, woody, floral, or sweet flavors [6,10,27,28] and played important roles in enhancing the aroma of BT. In contrast, the concentrations of VMs such as 2-methoxy-furan, *cis*-2-methyl-cyclopentanol, 1-hexanol, 9-methylheptadecane, 5-ethyl-5-methyl-decane, 3-methyl-tetradecane, and *O*-(1-naphthoyl)-1,2-benzenediol were significantly higher in the BoS (*p* < 0.05) than in the BT or STL samples.

To identify the differential VMs among the BT, STL, and BoS samples, we constructed the S-plot of the PLS-DA model by comparing the BT vs. STL, STL vs. BoS, and BT vs. BoS samples. In Figure 3a–c, we identified 28, 26, and 14 differential VMs when comparing the STL vs. BoS, BT vs. STL, and BT vs. BoS samples, respectively. As shown in Figure 3d, both common and unique metabolites were found among the different sample types paired for comparison. Specifically, the STL vs. BoS, BT vs. STL, and BT vs. BoS samples shared seven common substances, namely, 1-octen-3-ol, (E)-2-hexenal, 1-methyl-1H-1,2,4-triazole, (E)-3-hexen-1-ol, benzeneacetaldehyde, phenylethyl alcohol, and *trans*-β-ionone. The contents of these VMs were significantly different among the BT, STL, and BoS samples, which could be used as key VMs to distinguish tea samples and regulate aroma and flavor. Additionally, 19 common volatiles were found between STL vs. BoS and BT vs. STL, such as 2-ethyl-1-hexanol, 1-phenyl-1,2-propanediol, 3,6,6-trimethyl-bicyclo[3.1.1]hept-2-ene, 3-carene, isophorone, linalool, 4-chloro-2-methyl-1-phenyl-3-buten-1-ol, linalool oxide (pyranoid), 2-methoxy-3-(2-methylpropyl)-pyrazine, 3-hexenyl ester, (Z)-butanoic acid, 1-dodecanol, *n*-valeric acid *cis*-3-hexenyl ester, formic acid dodecyl ester, nerol, 6-methyl-pentadecane, and α-ionone. The contents of these VMs differed significantly between the STL and BT or BoS, and we used them as the key VMs on which the distinction of the glutinous rice aroma of STL was based. Furthermore, benzaldehyde and 1-hexanol were common substances between STL vs. BoS and BT vs. BoS; therefore, they were identified as the key VMs responsible for the improved BT aroma after the addition of STL. Finally, five unique substances were found to differ between the BT and BoS samples, namely, hexanal, *cis*-2-methyl-cyclopentanol, decane, 3,3,5-trimethyl-heptane, and (Z)-hexanoic acid-3-hexenyl ester. These compounds did not directly affect BT aroma with STL addition but were produced by indirect reactions and resulted in the detection of aroma differences between the BT and BoS.

#### 3.2.4. Analysis of Differential VMs

To further analyze the differential VMs among the BT, STL, and BoS samples, we constructed a thermography chart (Figure 4) based on 33 specific differential VMs (Appendix A). The BT and BoS samples with a sweet and glutinous rice aroma were clustered into one category, whereas the STL samples formed their own category exhibiting a strong glutinous rice aroma (Figure 4). In addition, the 33 VMs were grouped into three major classes. The first class included eight VMs, namely, (E)-2-hexenal, 1-methyl-1H-1,2,4-triazole, benzeneacetaldehyde, linalool, phenylethyl alcohol, and nerol, which had the highest contents in BT and distinctly contributed to the sweet aroma of the BT [5,23]. The second class included (E)-3-hexen-1-ol, *cis*-2-methyl-cyclopentanol, 1-hexanol, decane, and (Z)-hexanoic acid, 3-hexenyl ester, which were more abundant in the BoS samples than in either the BT or the STL samples; therefore, they were determined to be responsible for the sweet and glutinous rice aroma of BoS, following the combination of BT and STL. The third class included 20 VMs that were more abundant in the STL samples than in the other two samples; this category included compounds such as benzaldehyde, 1-octen-3-ol, 3-carene, 2-ethyl-1-hexanol, isophorone, linalool oxide (pyranoid), 2-methoxy-3-(2-methylpropyl)-pyrazine, 1-dodecanol, *n*-valeric acid *cis*-3-hexenyl ester, formic acid dodecyl ester, α-ionone, and *trans*-β-ionone, which contributed greatly to the strong glutinous rice aroma of the STL samples [7,12]. When combined with the Venn diagram, these findings clearly showed that the concentrations and proportions of the volatile substances have strong effects on the odor of each type of tea blend.

#### 3.2.5. OAV Analysis

OAV is widely used in the contribution evaluation of VMs to tea aroma; specifically, an OAV > 10 indicates a key contribution to odor composition, whereas an OAV > 1 indicates an important contribution to tea aroma [27,28,29]. We applied the 33 differential VMs (Appendix A), screened using PLS-DA/MEV/S-plot analysis, to the OAV analysis to identify the key VMs that determine the aroma quality in the BT, BoS, and STL samples. Combined with absolute sensory thresholds, the aroma characteristics and OAVs of 17 key VMs are listed in Table 3. There were 16 key differential volatiles with OAVs > 1, and the OAVs of benzaldehyde, 1-octen-3-ol, benzeneacetaldehyde, isophorone, 2-methoxy-3-(2-methylpropyl)-pyrazine, 1-dodecanol, α-ionone, and *trans*-β-ionone were above 500, indicating key contributions to the odor composition of the tea samples [28]. Among them, benzaldehyde and trans-β-ionone were confirmed as the major aroma contributors to sweet and floral odor in black tea [2,25,26], and their OAVs were 390/515 and 59942/177451 in BT/STL samples. The addition of STL significantly changed the volatile composition of black tea and directly affected the aroma quality. The OAVs of (E)-2-hexenal, hexanal, (E)-3-hexen-1-ol, linalool, and linalool oxide (pyranoid) were above 10, which also indicated strong contributions to tea odor composition (Table 3). (E)-3-hexen-1-ol, imparting a floral odor [23,28], with OAVs of 11/13/0.8, was identified in BT/BoS/STL samples. This indicated that the combination of BT and STL promoted the formation and enrichment of (E)-3-hexen-1-ol, which enhanced the aroma and flavor of BoS. Subsequently, 1-hexanol, phenylethyl alcohol, and nerol (OAV > 1) were identified as important contributors to sample aroma. When combined with the sensory evaluation results (Table 2), these results indicated that the OAVs of benzaldehyde, 1-octen-3-ol, isophorone, 2-methoxy-3-(2-methylpropyl)-pyrazine, linalool oxide (pyranoid), 1-dodecanol, α-ionone, and *trans*-β-ionone were the highest in the STL samples (*p* < 0.05), which were the key VMs of a glutinous rice aroma, and the OAVs of hexanal, (E)-2-hexenal, benzeneacetaldehyde, linalool, phenylethyl alcohol, and nerol were the highest in the BT samples, representing a sweet aroma. The combination of the BT and STL samples improved the contents and OAVs of hexanal, (E)-2-hexenal, (E)-3-hexen-1-ol, 1-hexanol, 1-octen-3-ol, benzeneacetaldehyde, linalool, phenylethyl alcohol, 2-methoxy-3-(2-methylpropyl)-pyrazine, 1-dodecanol, and nerol in the BoS samples, as they contributed distinctly to a sweet and glutinous rice aroma.

## 4. Conclusions

The addition of *Strobilanthes tonkinensis* Lindau (STL) during black tea (BT) processing had a substantial effect on the volatile metabolites (VMs) composition and aroma improvement of BT. We identified 141 VMs in STL firstly with high alcohol, ester, ketone, enyne, alkyne, and alkane contents. STL has a unique aroma of glutinous rice owing to high contents of benzaldehyde, 1-octen-3-ol, isophorone, 2-methoxy-3-(2-methylpropyl)-pyrazine, linalool oxide (pyranoid), 1-dodecanol, α-ionone, and *trans*-β-ionone in the STL leaves. Different addition methods and different addition concentrations of STL influence BT aroma and flavor considerably, and the addition of 20:1500 *w*/*w* STL to rolled tea leaves was the best method for processing BT with STL (i.e., BoS), with a strong and lasting glutinous sweet aroma and a sweet, mellow, and thick taste. PLS-DA clearly distinguished BT, STL, and BoS, and revealed 28, 26, and 14 differential volatile compounds when comparing STL vs. BoS, BT vs. STL, and BT vs. BoS, respectively. Combining S-plot, Venn, MEV, and OAV analysis, 16 key differential VMs were identified among BT, STL, and BoS. BoS combined the aroma properties of BT and STL and showed the highest contents of hexanal, (E)-2-hexenal, (E)-3-hexen-1-ol, 1-hexanol, 1-octen-3-ol, benzeneacetaldehyde, linalool, phenylethyl alcohol, 2-methoxy-3-(2-methylpropyl)-pyrazine, 1-dodecanol, and nerol. Our systematic study showed the effects of the new exogenous additive STL on the improvement of the BT aroma and value, thus providing sound theoretical support and insights for the development of new BT products as well as enriching our theoretical understanding of the mechanisms underlying aroma formation in BT. However, this study only shows the effects of STL addition on the VMs and aroma improvement of BT. Thus, the effect of STL addition on non-VMs and taste improvement requires further exploration.

## Figures and Tables

**Figure 1 foods-11-01678-f001:**
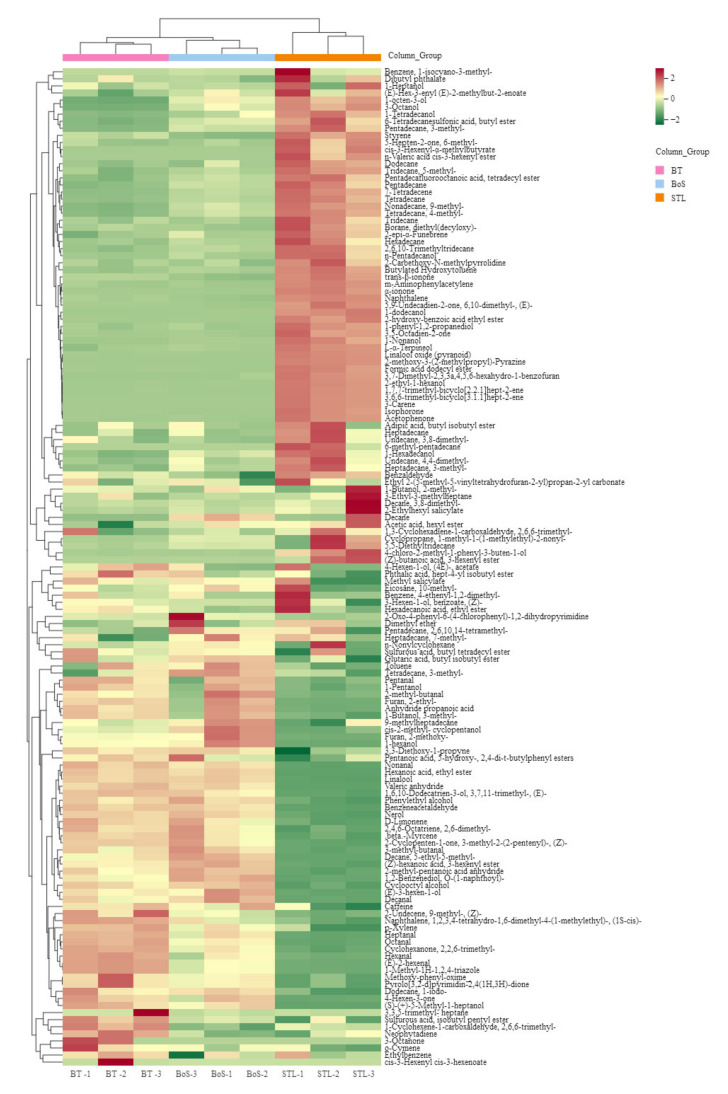
Component analysis showing the thermography of volatile metabolites in BT, STL, and BoS samples. STL, *Strobilanthes tonkinensis* Lindau; BT, black tea; BoS, black tea with added STL. Red represents high metabolite content, and green represents low metabolite content.

**Figure 2 foods-11-01678-f002:**
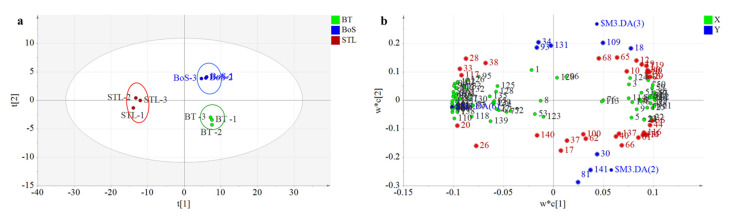
Partial least-squares discriminant analysis score plot (**a**) and loading plot (**b**) for volatile metabolites (VMs) in BT, STL, and BoS samples. STL, *Strobilanthes tonkinensis* Lindau; BT, black tea; BoS, black tea with added STL. SMI.DA(1), *Strobilanthes tonkinensis* Lindau (STL); SMI.DA(2), black tea (BT); SMI.DA(3), BT with added STL (BoS). Longitudinal data represent 141 VMs, and horizontal data represent three tea samples. Red represents high metabolite content, and blue represents low metabolite content.

**Figure 3 foods-11-01678-f003:**
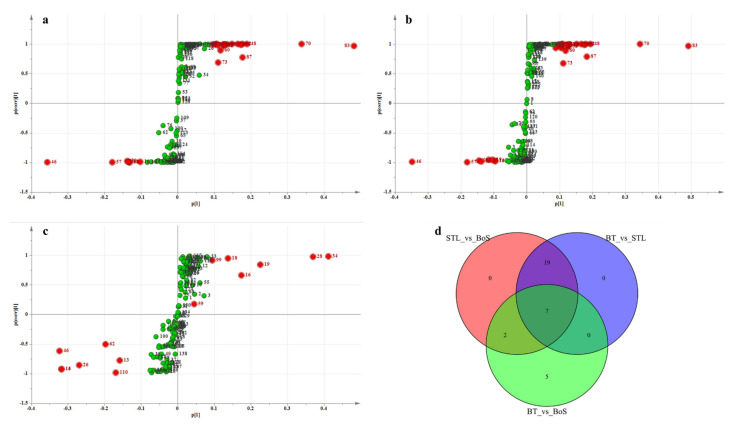
Differential volatile metabolites (VMs) in BT vs. STL, STL vs. BoS, and BT vs. BoS. S-plot generated from the PLS-DA model for BT vs. STL (**a**), STL vs. BoS (**b**), and BT vs. BoS (**c**). Green and red spots indicate all VMs involved in the model, and red spots show the most significant changes and higher contributions to the classification pattern. (**d**) Venn diagram of the differential VMs in BT vs. STL, BoS vs. STL, and BT vs. BoS. STL, *Strobilanthes tonkinensis* Lindau; BT, black tea; BoS, black tea with added STL.

**Figure 4 foods-11-01678-f004:**
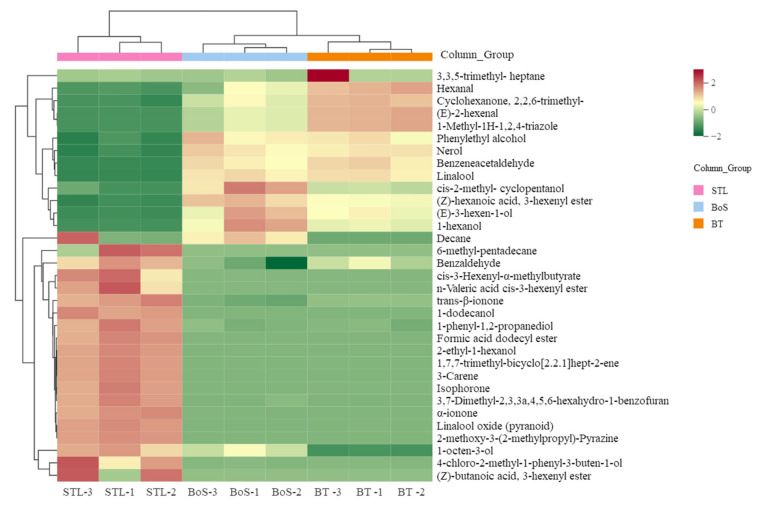
Component analysis showing the thermography of differential volatile metabolites in BT, STL, and BoS samples. STL, *Strobilanthes tonkinensis* Lindau; BT, black tea; BoS, black tea with added STL. Red represents high metabolite content, and green represents low metabolite content.

**Table 1 foods-11-01678-t001:** The detailed treatments of exogenously added *Strobilanthes tonkinensis* Lindau (STL) leaves during black tea processing.

Added Treatment	Types of Additives during BT Processing	Amounts of Additives (STL/Tea Leaves)
**#1**	STL/withered tea leaves	20/1500 g/g
**#2**	STL/rolled tea leaves	20/1500 g/g
**#3**	STL/fermented tea leaves	20/1500 g/g
**#4**	STL/rolled tea leaves	10/1500 g/g
**#5**	STL/rolled tea leaves	30/1500 g/g
**Blank control**	No additives	-

**Table 2 foods-11-01678-t002:** Sensory quality evaluations of tea samples treated with different addition methods of STL leaves.

STL Treatment	Appearance (25%)		Liquor Color (10%)		Aroma (25%)		Taste (30%)		Infused Leaf (10%)		Total Score
	Evaluation	Score	Evaluation	Score	Evaluation	Score	Evaluation	Score	Evaluation	Score	
**#1** (STL/withered tea leaves with 20/1500)	curly, tight, red, and glossy, uniform	90.0 ± 0.3 ^a^	red, clarity	90.2 ± 0.3 ^a^	strong glutinous rice flavor, covering tea aroma	87.0 ± 0.4 ^c^	strong glutinous rice taste, covering tea taste	86.8 ± 0.3 ^c^	red bright, stem with buds	89.9 ± 0.3 ^a^	88.3 ± 0.3 ^c^
**#2** (STL/rolled tea leaves with 20/1500)	curly, tight, red, and glossy, uniform	89.8 ± 0.3 ^a^	red, clarity	90.0 ± 0.2 ^a^	sweet, with glutinous rice flavor, strong, lasting	92.5 ± 0.3 ^a^	sweet mellow, thick, smooth	92.7 ± 0.2 ^a^	red bright, stem with buds	89.8 ± 0.2 ^a^	91.4 ± 0.2 ^a^
**#3** (STL/fermented tea leaves with 20/1500)	curly, tight, red, and glossy, uniform	90.1 ± 0.2 ^a^	red, clarity	89.9 ± 0.2 ^a^	sweet, with a little glutinous rice flavor	90.0 ± 0.4 ^b^	sweet mellow, a little thick	90.6 ± 0.3 ^b^	red bright, stem with buds	90.0 ± 0.4 ^a^	90.2 ± 0.3 ^b^
**#4** (STL/rolled tea leaves with 10/1500)	curly, tight, red, and glossy, uniform	90.0 ± 0.3 ^a^	red, clarity	90.0 ± 0.3 ^a^	sweet, with a little glutinous rice flavor	90.2 ± 0.3 ^b^	sweet mellow, a little thick	90.9 ± 0.3 ^b^	red bright, stem with buds	89.8 ± 0.2 ^a^	90.3 ± 0.3 ^b^
**#5** (STL/rolled tea leaves with 30/1500)	curly, tight, red, and glossy, uniform	89.9 ± 0.3 ^a^	red, clarity	89.8 ± 0.2 ^a^	strong glutinous rice flavor, covering tea aroma	86.7± 0.3 ^c^	strong glutinous rice taste, covering tea taste	86.3 ± 0.4 ^c^	red bright, stem with buds	89.9 ± 0.3 ^a^	88.0 ± 0.3 ^c^
**Blank control** (No additives)	curly, tight, red, and glossy, uniform	90.0 ± 0.2 ^a^	red, clarity	90.1 ± 0.3 ^b^	sweet, lasting	89.6 ± 0.5 ^b^	sweet mellow	90.3 ± 0.5 ^b^	red bright, stem with buds	89.8 ± 0.3 ^a^	90.0 ± 0.4 ^b^

Note: Data are presented as mean ± standard deviation (*n* = 5). Different lowercase letters in the same column indicate significant differences based on the least significant difference test (*p* < 0.05).

**Table 3 foods-11-01678-t003:** Aroma characteristics and odor activity value (OAV) of key differential volatile metabolites in BT, BoS, and STL.

No.	Name	OTs (µg/kg) ^A^	Aroma Characteristics ^B^	OAV		
				BT	BoS	STL
13	Hexanal	10	Fresh, green, fruity, sweaty	39.553 ± 2.031 ^a^	22.318 ± 8.438 ^b^	4.222 ± 0.743 ^c^
14	(E)-2-hexenal	13	Green, banana, fatty cheesy	93.353 ± 2.995 ^a^	50.848 ± 8.973 ^b^	3.267 ± 0.646 ^c^
16	(E)-3-hexen-1-ol	70	Green, floral, oily, earthy	11.061 ± 0.892 ^b^	13.756 ± 3.272 ^a^	0.823 ± 0.162 ^c^
19	1-hexanol	500	Ethereal, fruity, sweet, green	0.978 ± 0.038 ^b^	1.502 ± 0.383 ^a^	0.040 ± 0.010 ^c^
26	Benzaldehyde	3	Almond, caramel flavor	390.533 ± 34.965 ^b^	265.668 ± 77.463 ^c^	515.784 ± 36.185 ^a^
28	1-octen-3-ol	1	Mushroom, green, oily	169.18 ± 9.072 ^c^	836.963 ± 130.754 ^b^	1366.109 ± 150.229 ^a^
42	2-ethyl-1-hexanol	270,000	Citrus, fresh, floral, oily, sweet	0.000 ± 0.000 ^b^	0.000 ± 0.000 ^b^	0.008 ± 0.001 ^a^
46	Benzeneacetaldehyde	4	Woody, sweet, honey	2289.967 ± 158.374 ^a^	2125.484 ± 188.864 ^a^	139.284 ± 23.827 ^b^
48	Isophorone	2	Woody, sweet, green, tobacco	20.439 ± 0.611 ^b^	12.556 ± 0.807 ^c^	1211.616 ± 126.046 ^a^
57	Linalool	6	Citrus, floral, sweet, woody	374.445 ± 19.218 ^a^	371.961 ± 22.024 ^a^	15.894 ± 3.487 ^b^
59	Phenylethyl alcohol	750	Sweet, floral, fresh	2.380 ± 0.19 ^a^	2.516 ± 0.296 ^a^	0.615 ± 0.152 ^b^
70	Linalool oxide (pyranoid)	320	Floral, honey	0.600 ± 0.054 ^b^	0.545 ± 0.085 ^b^	24.518 ± 1.156 ^a^
72	2-methoxy-3-(2-methylpropyl)-pyrazine	0.015	Green, pea flavor, galbanum	186.988 ± 8.858 ^b^	205.408 ± 6.493 ^b^	40984.985 ± 1488.308 ^a^
75	1-dodecanol	0.066	Waxy, floral, honey, coconut	15.519 ± 13.684 ^b^	20.815 ± 18.117 ^b^	30780.489 ± 2020.076 ^a^
86	Nerol	300	Sweet, floral, neroli, magnolia	4.699 ± 0.146 ^a^	4.751 ± 0.452 ^a^	0.566 ± 0.235 ^b^
105	α-ionone	0.4	Sweet, woody, floral	71.301 ± 1.292 ^b^	67.817 ± 6.416 ^b^	1819.192 ± 135.53 ^a^
110	*trans*-β-ionone	0.007	Violet-like, floral, raspberry-like	59942.082 ± 444.755 ^b^	40619.259 ± 5716.759 ^c^	177451.466 ± 13572.469 ^a^

Note: BT, black tea; BoS, black tea and *Strobilanthes tonkinensis* Lindau; STL, *Strobilanthes tonkinensis* Lindau. Data were presented as mean ± standard deviation (*n* = 3). Different lowercase letters in the same column indicate significant differences based on the least significant difference test (*p* < 0.05). ^A^: OTs: odor thresholds in pure water. The OT values were according to the reported reference [20,25,26,27,28] or website: http://www.leffingwell.com/odorthre.htm (accessed on 1 February 2022). ^B^: The aroma characteristics of volatile metabolites were referred from http://www.thegoodscentscompany.com/search3.php (accessed on 1 February 2022).

## Data Availability

The data presented in this study are available in this article and Appendix A.

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
