# Peer review of "Effect of Strobilanthes tonkinensis Lindau Addition on Black Tea Flavor Quality and Volatile Metabolite Content"

_foods, 2022, doi:10.3390/foods11121678_

Round 1

Reviewer 1 Report

The author evaluated the effect of integrating the active component of STL in tea leaves on the conversion of volatile metabolites. The author employed a variety of statistical tools to support their findings, but the manuscript still needs revisions.

  1. Title : Why two titles are provided for the paper ?
  2. Table -1 : Experiment is not designed properly, its should be properly design to explain the effect of incorporation on tea quality.
  3. MEV analysis is not described in material and method section.
  4. Thermographic analysis techniques are clearly described in material method discussion.
  5. Difficult to interpret results from Fig-1. All the Figures should be self-explanatory.
  6. Conclusion: line no 37 : Author used different formulations not the different methods. So, the different methods should be avoided, also from other parts of the manuscript.
  7. Author should include more discussion on conversion of metabolites.

Author Response

Response to Reviewer 1 Comments

The author evaluated the effect of integrating the active component of STL in tea leaves on the conversion of volatile metabolites. The author employed a variety of statistical tools to support their findings, but the manuscript still needs revisions.

  1. Title : Why two titles are provided for the paper ?

Response: We apologize for this. The title has been revised, and only one title has been provided (line 2–3).

  1. Table -1 : Experiment is not designed properly, its should be properly design to explain the effect of incorporation on tea quality.

Response: Thank you for your suggestion. We have addressed this issue (line 99–100). Table 1 depicts the design of different processes and concentrations of Strobilanthes tonkinensis Lindau (STL) leaves during black tea processing. The follow-up detection, analysis, and research are to compare the differences among tea samples made through the design treatments in Table 1 to clarify the improvement of black tea quality by the addition of STL.

  1. MEV analysis is not described in material and method section.

Response: Thank you for your suggestion. In the revised manuscript, the description of MEV analysis has been added (line 154–156).

  1. Thermographic analysis techniques are clearly described in material method discussion.

Response: Thank you for mentioning this.

  1. Difficult to interpret results from Fig-1. All the Figures should be self-explanatory.

Response: Thank you for your suggestion. Fig. 1 has been revised (line 220).

  1. Conclusion: line no 37 : Author used different formulations not the different methods. So, the different methods should be avoided, also from other parts of the manuscript.

Response: There are two aspects of additions in this paper. One is different adding methods, that is, addition at withering, rolling, and fermentation stages. The second aspect is the different concentrations. The concluding section and the rest of the text reflect these aspects.

  1. Author should include more discussion on conversion of metabolites.

Response: Thank you for your suggestion. In the revised manuscript, the discussion of metabolite transformation has been appropriately added (lines 382–403). STL is a dry leaf state, which has obvious aroma and taste attributes. Its addition has little impact on the internal aroma and taste metabolism of tea itself. The changes in aroma components, concentration, and proportion mainly impact the aroma of black tea.

Reviewer 2 Report

Dear author(s):

Effect of Strobilanthes tonkinensis Lindau addition on black tea flavor quality and volatile metabolite content

After an exhaustive revision, the manuscript is Reconsider after major revision (control missing in some experiments). In general, the study is closely connected to the journal's objectives. The study is very interesting. The English is good. The introduction is complete, very detailed, but it needs to add references from 2022. The section results and discussion should be improved in some points, since it present very complete description of the results, but the authors need to add a lot of information on explication of the results, comparison with other studies, and explication (discussion) of the results obtained with respect to other studies. In the following pages, I give a detailed revision of the manuscript.

ABSTRACT

The abstract is good. However, the authors need to add numerical results.

  1. INTRODUCTION

The introduction is very clear, concise and precise, with good English, and it has updated references until 2021. The authors need to add references (some lines) from 2022. I recommend these references to add in the first lines of the introduction:

  1. MATERIALS AND METHODS

General comments

This section is clear. The English is good. The authors must add a Figure that represents all the complete methodology. This Figure will help to understand the methodology.

2.2. Tea processing and sample treatment

What are the references for each step?

2.3. Tea sensory quality evaluation

Why 5 panelists? Is it a little amount of panelists?

  1. RESULTS AND DISCUSSION

The section of “Results and Discussion” is characterized by a very detailed description of the results, explication of the results, comparison with other studies, and explication (discussion) of the results obtained with respect to other studies.

3.1. Effects of STL additions on the sensory qualities of BT

The authors present a very complete description of the results, explication of the results, and comparison with other studies. However, the authors need to add explication (discussion) of the results obtained with respect to other studies.

3.2. Effects of STL addition on BT VMs

3.2.1. Identification of VMs

The authors present a very complete description of the results. However, the explication of the results is weak, i.e., the authors need to add more lines. The authors mention a comparison with other studies, but the authors need to add explication (discussion) of the results obtained with respect to other studies.

3.2.2. MEV analysis

The authors present a very complete description of the results. However, the authors need to add the explication of the results for all the observations, i.e., the authors need to add more lines. The authors mention a comparison with other studies, but the authors need to add explication (discussion) of the results obtained with respect to other studies.

3.2.3. PLS-DA analysis

The authors present a very complete description of the results. However, the authors need to add the explication of the results, the comparison with other studies, and the explication (discussion) of the results obtained with respect to other studies.

3.2.4. Analysis of differential VMs

The authors present a very complete description of the results. However, the authors need to add the explication of the results, the comparison with other studies, and the explication (discussion) of the results obtained with respect to other studies.

3.2.5. OAV analysis

The authors present a very complete description of the results, with few lines on explication of the results, and the authors need to add the comparison with other studies, and the explication (discussion) of the results obtained with respect to other studies.

  1. CONCLUSIONS

The authors should add the section conclusions.

Author Response

Response to Reviewer 2 Comments

Dear author(s):

Effect of Strobilanthes tonkinensis Lindau addition on black tea flavor quality and volatile metabolite content

After an exhaustive revision, the manuscript is Reconsider after major revision (control missing in some experiments). In general, the study is closely connected to the journal’s objectives. The study is very interesting. The English is good. The introduction is complete, very detailed, but it needs to add references from 2022. The section results and discussion should be improved in some points, since it present very complete description of the results, but the authors need to add a lot of information on explication of the results, comparison with other studies, and explication (discussion) of the results obtained with respect to other studies. In the following pages, I give a detailed revision of the manuscript.

ABSTRACT

The abstract is good. However, the authors need to add numerical results.

Response: Thank you for your suggestion. In the revised manuscript, the numerical results have been appropriately added (lines 9–23).

  1. INTRODUCTION

The introduction is very clear, concise and precise, with good English, and it has updated references until 2021. The authors need to add references (some lines) from 2022. I recommend these references to add in the first lines of the introduction:

Response: Thank you for your valuable suggestion. In the revised manuscript, references from 2022 have been added (references 2, 25, and 28).

  1. MATERIALS AND METHODS

General comments

This section is clear. The English is good. The authors must add a Figure that represents all the complete methodology. This Figure will help to understand the methodology.

Response: Thank you for your suggestion. The methodologies used in this article were as follows: tea processing, sensory quality evaluation, volatile metabolite analysis, and statistical analysis, which have been clearly described in our manuscript. We have followed the pattern observed in most articles on similar topics. Therefore, we have not added a figure there. We hope you agree with this.

2.2. Tea processing and sample treatment

What are the references for each step?

 Response: Thank you for pointing this out. In the revised manuscript, the reference has been added (lines 73–74).

2.3. Tea sensory quality evaluation

Why 5 panelists? Is it a little amount of panelists?

Response: This is set according to China’s tea sensory evaluation standards. We followed the number of evaluation experts typically selected for flavor evaluation.

  1. RESULTS AND DISCUSSION

The section of “Results and Discussion” is characterized by a very detailed description of the results, explication of the results, comparison with other studies, and explication (discussion) of the results obtained with respect to other studies.

3.1. Effects of STL additions on the sensory qualities of BT

The authors present a very complete description of the results, explication of the results, and comparison with other studies. However, the authors need to add explication (discussion) of the results obtained with respect to other studies.

Response: Thank you for your suggestion. A detailed description and discussion of the results obtained with respect to other studies has been added. Because there is no previous research on the effect of STL addition on tea flavor, there is limited analysis that can be used for comparison (lines 165–167, 182–184).

3.2. Effects of STL addition on BT VMs

3.2.1. Identification of VMs

The authors present a very complete description of the results. However, the explication of the results is weak, i.e., the authors need to add more lines. The authors mention a comparison with other studies, but the authors need to add explication (discussion) of the results obtained with respect to other studies.

Response: Thank you for your suggestion. The analysis of the results and the explication (discussion) of the results obtained with respect to other studies have been appropriately added (lines 193–195, 208–211).

3.2.2. MEV analysis

The authors present a very complete description of the results. However, the authors need to add the explication of the results for all the observations, i.e., the authors need to add more lines. The authors mention a comparison with other studies, but the authors need to add explication (discussion) of the results obtained with respect to other studies.

Response: Thank you for your suggestion. The analysis of the results and the explication (discussion) of the results obtained with respect to other studies have been appropriately added (lines 231–233, 240–241).

3.2.3. PLS-DA analysis

The authors present a very complete description of the results. However, the authors need to add the explication of the results, the comparison with other studies, and the explication (discussion) of the results obtained with respect to other studies.

Response: Thank you for your suggestion. We have addressed this by adding the analysis of the results and interpretation with respect to other studies (lines 272–281, 289–291).

3.2.4. Analysis of differential VMs

The authors present a very complete description of the results. However, the authors need to add the explication of the results, the comparison with other studies, and the explication (discussion) of the results obtained with respect to other studies.

Response: We agree with this suggestion. We have included the analysis of the results and the explication (discussion) of the results obtained with respect to other studies (lines 3210–323, 326–333).

3.2.5. OAV analysis

The authors present a very complete description of the results, with few lines on explication of the results, and the authors need to add the comparison with other studies, and the explication (discussion) of the results obtained with respect to other studies.

Response: We have added the necessary discussion and explanations as indicated (lines 350–359).

  1. CONCLUSIONS

The authors should add the section conclusions.

Response: We apologize for this gap in the earlier version. The conclusion has been appropriately added (lines 382–403).

Round 2

Reviewer 1 Report

Author did significant changes in the revised manuscript,  Author is requested to format the Table-2 properly.  

Author Response

Response: Thank you for your suggestion. In the revised manuscript, Table 2 has been revised to clearly depict the sensory quality evaluations results of tea samples treated with different addition methods and concentrations of STL leaves. (lines 178-179).

Reviewer 2 Report

Dear Author(s)

The resubmitted manuscript has been completely improved compared to its previous version.

Best regards

Author Response

Response: Thank you for your valuable suggestion. Best regards.